# The Regulation of ROS and Phytohormones in Balancing Crop Yield and Salt Tolerance

**DOI:** 10.3390/antiox14010063

**Published:** 2025-01-07

**Authors:** Lei Jiang, Minggang Xiao, Rongfeng Huang, Juan Wang

**Affiliations:** 1Biotechnology Research Institute, Chinese Academy of Agricultural Sciences, Beijing 100081, China; jianglei031197@163.com (L.J.); rfhuang@caas.cn (R.H.); 2Biotechnology Research Institute, Heilongjiang Academy of Agricultural Sciences, Harbin 150028, China; xiaominggang@haas.cn

**Keywords:** reactive oxygen species signaling, oxidative stress, phytohormone, salt stress, crop yield, superior alleles, molecular breeding

## Abstract

Salinity affects crop growth and productivity, and this stress can be increased along with drought or high temperature stresses and poor irrigation management. Cultivation of salt-tolerant crops plays a critical role in enhancing crop yield under salt stress. In the past few decades, the mechanisms of plant adaptation to salt stress have been described, especially relying on ionic homeostasis, reactive oxygen species (ROS) scavenging, and phytohormone signaling. The studies of these molecular mechanisms have provided a basis for breeding new salt-tolerant crop germplasm and have facilitated the entry into the era of molecular breeding of salt-tolerant crops. In this review, we outline the recent progress in the molecular regulations underlying crop salt tolerance, focusing on the double-edged sword effect of ROS, the regulatory role of phytohormones, and the trade-off effects of ROS and phytohormones between crop yield and salt tolerance. A future challenge is to identify superior alleles of key salt-tolerant genes that will accelerate the breeding of high-yield and salt-tolerant varieties.

## 1. Introduction

The area of salt-affected land worldwide is approximately 950 million hectares and is still increasing, which seriously threatens the production of crops [1,2]. As an important reserve of cultivated land resources, the productivity of salt-affected land urgently needs to be enhanced. Therefore, it is of practical significance to improve the salt tolerance of crops and cultivate high-yield salt-tolerant crops. The salt-tolerant varieties generally have no advantage in terms of yield. Generally speaking, the rice variety with the strongest salt-stress tolerance displays the lowest grain yield, and this contradiction restricts the breeding process of high-yield salt-tolerant crops [3].

When plants are exposed to salt stress, the excessive accumulation of Na^+^ disrupts the Na^+^/K^+^ homeostasis within plant cells, resulting in the inhibition of numerous crucial physiological processes [4]. During the adaptation to salt stress, plants have evolved a series of signaling pathways to deal with salt stress. The salt overly sensitive (SOS) signaling pathway is composed of the core components Na^+^/H^+^ antiporter SOS1, serine/threonine protein kinase SOS2, and calcium-binding protein SOS3/SCABP8 [5,6]. Recent studies have indicated that reducing high-affinity K^+^ transporters (HKTs) and increasing vacuolar Na^+^/H^+^ exchanger 1 (NHX1) and the CHX-type ion transporter 1 (CHX1) can jointly enhance Na^+^ efflux, restrict Na^+^ uptake, and sequester Na^+^ in vacuoles to alleviate the damage caused by salt stress [7,8,9].

Salt stress not only induces ionic stress, but also causes the excessive accumulation of ROS, such as hydrogen peroxide (H_2_O_2_), superoxide radical (O_2_^•^¯), singlet oxygen (^1^O_2_), and hydroxyl radical (OH•), resulting in oxidative stress and nutrient deficiency. ROS are products under stresses and signaling molecules for growth and development, and ROS homeostasis is of great significance to plants [10]. Accumulating research has elucidated the signaling, generation, and metabolism of ROS [10,11,12]. Phytohormones are small chemical substances that play a vital role in the growth and development of plants. Moreover, phytohormones also play significant roles when plants cope with stresses. Increased or decreased levels of certain hormones could help plants adapt to salt stress, including abscisic acid (ABA), ethylene (ET), jasmonic acid (JA), salicylic acid (SA), auxin, cytokinin (CK), gibberellic acid (GA), and brassinosteroid (BR) [13]. Whether to regulate the normal growth of plants or mediate the responses to abiotic stress, the integration and coordination of multiple plant hormones are requisites for plants. Moreover, ROS homeostasis and phytohormone signaling pathways are interrelated and form a more complex regulatory network in plants [10,13], revealing the potential functions of ROS and phytohormones in plant growth and salt-stress responses, probably responsible for the key modulation of the collaborative balance between crop yield and salt tolerance. By searching published research in the National Center for Biotechnology Information (NCBI) PubMed https://www.ncbi.nlm.nih.gov/ (accessed on 1 October 2024) database using “ROS or hormones”, “salt stress”, and “crop yield” as key words, a tremendous number of reports were found, followed by a selection focusing on crops (rice, wheat, maize, soybean, potato, etc.). Here, we mainly review the latest research progress in the regulation of ROS homeostasis and phytohormones in crops under salt stress, especially the critical genes in the last ten years.

## 2. ROS Homeostasis Under Salt Stress

### 2.1. Signaling Function of ROS Under Salt Stress

ROS have important roles throughout the entire life cycle of plants. Under normal circumstances, ROS are maintained at a relatively low level and can serve as signaling molecules to regulate plant growth and development [14,15]. In the early stage of salt stress, ROS are produced “actively” or “passively” in plant cells [10]. The extracellular ROS burst is mediated by plasma-membrane-localized respiratory burst oxidase homologs (RBOHs), which utilize NADPH in the cytosol to generate O_2_^•^¯ in the apoplast and convert it into H_2_O_2_ through the superoxide dismutase [16]. The RBOH family in rice (*Oryza sativa*) contains nine members (*OsRbohA* to *I*) [17]. Research has disclosed that not every RBOH serves as the principal protein for ROS bursts in response to salt stress. OsRBOHA and OsRBOHI are the key factors in rice, whereas AtRBOHD and AtRBOHF are the crucial ones in *Arabidopsis* (*Arabidopsis thaliana*) [16,17]. The H_2_O_2_ generated by RBOHs is capable of mediating calcium ion signaling in peas, further activating RBOHD by binding to the EF-hand domain of RBOHD, thereby augmenting H_2_O_2_ production [18,19]. It has been discovered that H_2_O_2_ induced by salt stress in soybeans (*Glycine max*) promotes oxidative modification of the transcription factor GmNTL1 protein, facilitating its translocation to the nucleus to activate the expression of *GmRbohBs*, further enhancing the ROS level and thereby rapidly amplifying the ROS signal elicited by salt stress [20]. It has been reported that the trafficking of RBOHs is regulated by the epsin-like clathrin adaptor 4 (ECA4) [21]. Furthermore, RBOHs are also regulated by ubiquitylation, persulfidation, nitrosylation, and endocytosis [10].

Apart from the extracellular space, ROS are also produced in various cellular compartments, such as mitochondria, the cytosol, peroxisomes, and the endoplasmic reticulum [12,22]. Salt stress could also damage the ultrastructure of chloroplasts, resulting in the generation of H_2_O_2_ and O_2_^•^¯ in chloroplasts [23]. Due to the fact that aquaporins (AQPs) identified on the plasma membrane and organelle membranes promote the bidirectional transportation of H_2_O_2_, the H_2_O_2_ level within the cytoplasm can influence the H_2_O_2_ level in other cellular compartments, and vice versa [24].

During the process of salt stress, the interplay between the generation and scavenging of ROS in each cellular compartment gives rise to compartment-specific ROS signals [10]. These signals, in combination with other non-ROS signals, traverse the cytoplasm to reach the nucleus, modify nuclear ROS signals, and trigger defense and adaptation responses [25]. It has been discovered that ROS could regulate gene expression through mRNA stability and microRNA abundance during salt-stress responses. For instance, H_2_O_2_ could enhance the stability of SOS1 mRNA to facilitate the Na^+^ efflux [26]. Stress-induced ROS inhibit the accumulation of *miR408* in maize (*Zea mays*), thereby elevating the transcription levels of its target genes *ZmLAC9* and *ZmLAC18* and promoting cell wall development and salt tolerance by regulating the polymerization of lignin monomers [27]. In addition to regulating transcription via epigenetic mechanisms during stress, ROS also influence the translocation of different redox-regulated transcriptional regulators. Non-expressor of pathogenesis-related genes 1 (NPR1), heat shock factors (HSFs), and membrane-bound NAC transcription factors have been determined to sense H_2_O_2_ signals in the cytoplasm and convey them to the nucleus to modulate the expression of salt-stress response genes [12,28,29]. These studies indicate that ROS signals play a crucial role in the salt-stress response.

### 2.2. Oxidative Stress Induced by ROS Under Salt Stress

Appropriate levels of ROS constitute an essential signaling molecule for activating salt stress tolerance; however, excessive accumulation of ROS could elicit oxidative stress in plants, disrupting the oxidation–reduction balance of cells and resulting in DNA damage, protein degradation, as well as peroxidation of carbohydrates and lipids [30]. At present, there have been detailed reviews discussing the molecular mechanisms of ROS scavenging under oxidative stress [12]. To counteract oxidative stress, there is a large number of antioxidant enzymes that clear ROS [31]. Enzymatic scavengers include superoxide dismutase (SOD), catalase (CAT), ascorbic acid peroxidase (APX), glutathione peroxidase (GPX), glutathione reductase (GR), dehydroascorbate reductase (DHAR), monodehydroascorbate reductase (MDHAR), and glutathione S-transferase (GST) [32,33,34]. In an S-type cytoplasmic male sterile maize line, the moderately expressed sterility gene open reading frame 355 (ZmORF355) in the mitochondria promotes salt tolerance by activating various antioxidant enzymes [35]. Biochemical analyses have revealed that hydroquinone could be transformed into arbutin under the catalysis of UDP-glucose transferases (UGTs). Moreover, the growth regulating factor 7 (OsGRF7) could directly bind to the promoters of OsUGT1 and OsUGT5 to stimulate their expression, thereby increasing arbutin biosynthesis and enlarging grains [36]. Inhibition of *ZmmiR169q* under salt stress contributes to salt tolerance by mediating ROS elimination, by increasing the expression of its target gene ZmNF-YA8 and enhancing the transcription of the antioxidant enzyme gene peroxidase 1 (ZmPER1) [37]. The salt-tolerant receptor-like cytoplasmic kinase 1 (STRK1) phosphorylates and activates catalase C (CatC), promoting the homeostasis of H_2_O_2_ in rice. Plants overexpressing OsSTRK1 exhibit a higher catalase activity, a lower H_2_O_2_ accumulation, and a greater salt tolerance compared to wild-type plants [38]. Further research has found that catalase 1 phosphatase (PC1)-mediated dephosphorylation of CAT can inhibit its activity [39]. These studies indicate that the activity regulation of many antioxidant enzymes under salt stress is a key regulatory node.

In addition to antioxidant enzymes, an increasing number of studies indicate that many non-enzymatic antioxidants are involved in mitigating ROS-induced damage under salt stress, such as ascorbic acid (AsA), anthocyanin, glutathione, and tocopherols [2]. It has been discovered that AsA confers salt tolerance to plants [40]. GDP-mannose pyrophosphorylase VTC1 is a key enzyme in the AsA biosynthesis pathway and positively regulates salt tolerance [41,42,43]. OsVTC1-1 plays a crucial role in rice salt tolerance by promoting the generation of AsA to scavenge excessive ROS [44]. Maize bronze 1 (ZmBZ1) enhances stress tolerance by mediating anthocyanin accumulation [45]. Branched-chain amino transferase 2 (OsBCAT2) governs the catabolism of branched-chain amino acids (BCAAs) and the synthesis of pantothenic acid (vitamin B5, VB5) in rice. Salt stress induces the expression of *OsBCAT2*, accelerating the intracellular degradation of BCAAs and augmenting the synthesis of the downstream product VB5, thereby enhancing the salt stress tolerance of rice [46]. *OsGSA1* encodes a UDP-glucosyltransferase, enabling plants to accumulate flavonoid glycosides, thereby protecting rice from abiotic stress [47], revealing that many antioxidants play a role in crop salt tolerance.

Thus, the studies in recent years have emphasized the significant role of ROS homeostasis in the response to salt stress. ROS show the double-edged sword effects in different crops in terms of their contribution to salt tolerance (Table 1). To prevent damage, it is essential for plants to strictly control ROS concentrations. Enhancing crop salt tolerance by recruiting enzymatic and non-enzymatic antioxidant systems to eliminate excessive ROS is an effective approach (Figure 1).

## 3. Modulation of Phytohormones in Salt-Stress Responses

The phytohormones facilitate plants’ rapid responses to salt stress. Some hormones exert positive regulatory effects on plant salt tolerance, such as ABA and JA, whereas some have negative regulatory roles, such as GA [13]. Moreover, the regulatory function of some hormones in salt tolerance is opposite in different species. For instance, ethylene has a positive role in salt tolerance in *Arabidopsis*, while it negatively regulates salt tolerance in rice [51,52,53], demonstrating that phytohormones have complicated regulatory mechanisms.

When plants are exposed to salt stress, the accumulation level of ABA shows a positive correlation with the salt tolerance of plants [54,55]. ABA could activate a variety of downstream signaling factors, such as kinases, phosphatases, and G proteins. It has been discovered that the drought-induced protein GmDi19-5 could negatively regulate the responses of soybeans to drought and salt stress through the ABA-dependent pathway [56,57]. In rice, the transcription factor OsSGL is involved in inhibiting the gene expression of *OsNCED3*, encoding a key enzyme of ABA biosynthesis. Under salt stress, OsSGL undergoes acetylation modification and nuclear–cytoplasmic distributions, enabling a dynamic regulation of ABA content in plant cells [58,59]. Consequently, ABA, as a stress hormone, plays an important role in regulating salt tolerance in crops.

It has been discovered that ethylene treatments augment the salt sensitivity of rice seedlings, while, for *Arabidopsis* and soybeans, they enhance salt tolerance [51,60,61]. Salt intolerance 1 (OsSIT1), a lectin receptor-like kinase and an ethylene sensor, actively mediates salt sensitivity by activating the mitogen-activated protein kinase 3/6 (MPK3/6), thereby promoting ethylene biosynthesis, ROS accumulation, and sensitivity to salt stress in rice [62].

GA plays a crucial role in modulating growth under abiotic stress [63]. The genes involved in the GA catabolic pathway, i.e., gibberellin 2-oxidase 5/7 (*GA2ox5/7*), *OsMYB91*, and *OsCYP71D8L*, contribute to decreased GA levels and a retarded growth, thereby enhancing the salt tolerance of plants [64,65,66]. Moreover, OsDSK2a, functioning as a ubiquitin receptor and transporter, promotes the degradation of the GA-inactivating enzyme EUI in rice. The level of OsDSK2a is inhibited by salt stress, giving rise to the accumulation of EUI and the reduced level of biologically active GA, elucidating the mechanism underlying the inhibition of salt stress in the context of plant growth [67]. These studies indicate that growth inhibition under salt stress might be an active adaptive mechanism in crops.

JA plays a key role in plant growth and development, in the defense against insects and pathogens, and in secondary metabolism. It has also been unveiled that JA possesses significant functions in plant responses to salt stress [68]. When plants encounter high-concentration salt stress, degradation of the JA response factor JAZ8 relieves its inhibitory effect on the NF-YA1-YB2-YC9 transcription factor complex, promoting the expression of salt-stress response genes and enhancing salt tolerance in rice [69,70,71]. IbNAC087, a nuclear-localized transcriptional activator in sweet potatoes (*Ipomoea batatas*), could directly activate the expression of JA synthesis-related genes to increase JA content and, thus, enhance salt and drought resistance [72].

It has been found that the mutation of rice salt tolerance 1 (*RST1*) in rice, encoding the auxin response factor OsARF18, promotes nitrogen utilization and reduces the accumulation of NH4^+^ under salt stress, thereby enhancing salt tolerance and grain yield [73]. Overexpression of rice big grain 1 (RBG1) leads to an increased grain size due to the elevated cell numbers by increasing auxin accumulation; meanwhile, RBG1 could upregulate stress proteins to enhance the tolerance to the heat and osmotic and salt stresses in rice [74,75]. Therefore, promoting growth to a certain extent is also necessary for improving salt tolerance in crops.

Some reports have indicated that cytokinin regulates salt tolerance in *Arabidopsis* [76,77,78]. Furthermore, a study on maize has determined that a type A response regulator, ZmRR1, modulates Cl^−^ exclusion and negatively regulates cytokinin signaling. ZmRR1 could interact with a key mediator of cytokinin signaling, ZmHP2, to inhibit Cl^−^ exclusion and salt tolerance [79]. This study demonstrates that cytokinin signaling has a positive role in salt tolerance through ionic homeostasis in crops.

Taken together, the roles of phytohormones in crop growth and development under salt stress are complicated (Table 2), and more in-depth research on some hormones in crops is needed. In particular, the hormones mainly regulating plant growth have a promising route for balancing salt tolerance and crop yield.

## 4. ROS and Phytohormones Jointly Balance Crop Yield and Salt Tolerance

ROS play a dual role in plant growth and stress responses, and phytohormones are also significant endogenous chemical signals that orchestrate plant growth and development in appropriate environments and under adverse conditions [10,13]. Owing to the high reactivity and oxidizing nature of ROS, ROS signaling is predominantly mediated by protein oxidative post-translational modifications. This characteristic enables ROS to target a broad spectrum of proteins and to dynamically and spatiotemporally regulate signaling transduction pathways under various stress conditions, a phenomenon which is similar to phytohormone signaling pathways [10]. It has been reported that phytohormones could be associated with ROS to jointly regulate plant growth, development, and adaptation to adverse circumstances.

First and foremost, phytohormones are capable of influencing the synthesis, metabolism, and signaling of ROS. Research has discovered that wheat (*Triticum aestivum*) brassinazole-resistant 1 (TaBZR1) could not only directly bind to the G-box motif in the promoter of the key ABA synthesis gene *TaNCED3* to activate its expression and ABA synthesis, but also bind to the promoters of the ROS scavenging-related genes *TaGPX2* and *TaGPX3* to promote their expression, thereby facilitating ROS scavenging [80]. It is interesting that ethylene inhibits but ABA promotes the accumulation of ROS in *Arabidopsis* [81]. Furthermore, AsA acts as a key factor in integrating the interaction of ethylene and ABA in the regulation of ROS levels. Ethylene and ABA antagonistically regulate AsA biosynthesis via ethylene-insensitive 3 (EIN3) and ABA insensitive 4 (ABI4), respectively, both of which are key factors in the ethylene and ABA signaling pathways, respectively [81]. In addition, EIN3 and ABI4 collaboratively modulate salt tolerance via AsA biosynthesis [82]. It is noteworthy that the GA signaling factor RGA1 (alpha subunit of heterotrimeric G protein) in rice negatively regulates salt tolerance by participating in ROS scavenging by modulating antioxidant enzymes protein abundance [83]. The most recent study has disclosed that, under salt stress, the downstream transcription factor OsMYB2 of ABA mediates the expression of the amino acid transporter gene ANT1, further enhancing the transportation of amino acids (Pro, Leu, Phe, Tyr) and regulating the balance between rice growth and salt tolerance through the accumulation of amino acids like Pro and Tyr and the reduction of ROS. This research provides novel insights into the function of plant amino acid transporter genes [84].

Secondly, ROS exert an influence on the synthesis, metabolism, and signaling of phytohormones. It has been discovered that GPX3 is not merely a ROS-scavenging enzyme highly sensitive to H_2_O_2_, and that the redox state of GPX3 could also influence the phosphatase activity of its interacting partner ABA insensitive 2 (ABI2), which is a negative component of the ABA signaling pathway and participates in stomatal closure [85]. H_2_O_2_ could sulfonate BZR1 to enhance its transcriptional activity, thereby positively regulating BR signaling and plant growth in *Arabidopsis* [86]. It is notable that H_2_O_2_ could affect both the contents of IAA and ABA through the tryptophan synthase B subunit 1 (TSB1). Salt-induced sulfoxidation of TSB1 increases ABA accumulation and facilitates salt stress tolerance in *Arabidopsis* [87]. Thus, ROS-mediated oxidative modification of phytohormone signaling factors confers to plants the ability to adapt to salt stress.

Furthermore, there exist certain transcription factors that could concurrently regulate both ROS and phytohormones. The mutation of the key regulatory factor ITPK4 in *Arabidopsis* leads to an augmented accumulation of ROS and decreased auxin levels in plants under salt stress [88]. In soybeans, the induced expression of *GmSIN1* under salt stress upregulates the cellular levels of ABA and ROS, giving rise to an amplification of the initial salt stress signal [89]. In brief, plants utilize multiple signaling molecules to coordinate developmental processes and responses to environmental stimuli [90] (Table 3).

In recent years, numerous reviews related to balancing crop growth and stress tolerance have been reported [3,91]. Regarding the balance between crop yield and salt tolerance, it seems to be distinct from other stresses, such as drought, heat, and cold, since the salt content in saline–alkali soil changes with the weather [91]. Recent studies have discovered that, in natural populations of wheat, there are many natural variations of the promoter of *TaCHP*, forming two main haplotypes, Hap1 and Hap2. The salt tolerance of the varieties carrying Hap1 is significantly higher than that of thosse carrying Hap2. TaCHP-Hap1 could stably increase the yield of wheat in saline–alkali soil without affecting the yield in normal soil. Therefore, the salt-tolerant haplotype of *TaCHP* has a strong application potential in molecular design breeding of wheat [48]. Under saline–alkaline conditions, high pH stress amplifies the harmful effects of salt stress on plants. Recent studies have shown that the *Alkaline tolerance 1* (*AT1*) gene in *Sorghum* negatively regulates crop tolerance to alkali by modulating ROS homeostasis [49,50]. It has been determined that *OsAT1* (known as grain size 3, OsGS3) is a high-yield gene that encodes an atypical G protein γ subunit [92]. Introducing the favorable allele of *OsAT1* into the high-quality rice variety Kongyu131 could increase the yield by 27.8% under alkaline stress conditions, indicating that this allele has a great potential value in cultivating alkali-tolerant rice [50]. Recent research has revealed that OsDSK2a, a co-regulatory factor for growth and salt tolerance in rice, could be phosphorylated by the conserved kinase SnRK1A, and natural variations exist at the phosphorylation sites. The varieties contained in the elite haplotype *OsDSK2a-G* have a higher grain yield than that of the varieties contained in *OsDSK2a-S* under salt stress [93]. Furthermore, a study has reported, for the first time, a grain length gene GL12 from the common wild rice (*Oryza rufipogon*) which is capable of concurrently enhancing the grain length and salt tolerance of rice. A G/T variation site in its promoter region influences its spatio-temporal expression pattern in the spikelets and roots, thereby balancing the yield and salt tolerance of rice [94]. Therefore, it is still necessary to explore the key salt-tolerant genes in crop plants, regardless of whether these genes are related to ROS or hormone pathways, and try to identify the associated excellent natural variations as much as possible to provide opportunities for the trade-off between salt tolerance and growth (Figure 2).

## 5. Conclusions and Future Prospects

Crop yield has been greatly improved after years of breeding, but the improvement in crop salt tolerance is still at a bottleneck. The identification of key salt-tolerant genes in plants and an in-depth exploration of their molecular regulatory mechanisms could provide significant theoretical foundations for the breeding of salt-tolerant crops. However, crop salt tolerance is a quantitative trait controlled by multiple genes. Due to the subtle effect of one or several genes, more genes need to work together to achieve synergistic effects under salt stress. Therefore, highly efficient methods for identifying salt-tolerant genes are a challenge for molecular breeding. Meanwhile, with the development of functional genomics and multiomics technology, a stable phenome identification system under salt stress is very important.

On the other hand, since the modulations of ROS and phytohormones in the context of plant growth and salt stress have been deeply researched in model plants, it is feasible to use these regulations to balance yield and salt tolerance in crops. Phytohormones could play a significant regulatory role in extremely small amounts, providing a potential way for balancing ROS homeostasis, salt tolerance, plant growth, and yield enhancement. GA, which was reduced to increase the harvest index in the Green Revolution, could also improve salt tolerance to a certain level. Identification of key genes referring to the crosstalk between ROS and phytohormones is helpful to control their dose effects.

During the prolonged breeding process, with yield traits as the main target orientation, the genetic diversity of crops has decreased, and superior alleles for salt tolerance have been lost or lacked directed selection. The discovery of superior alleles that can collaboratively improve both yield and salt tolerance remains an exciting challenge. To increase the productivity of salt–alkali land and improve grain yield, it is important to apply modern molecular breeding approaches for precise, rapid, and efficient improvement of crop salt tolerance.

## Figures and Tables

**Figure 1 antioxidants-14-00063-f001:**
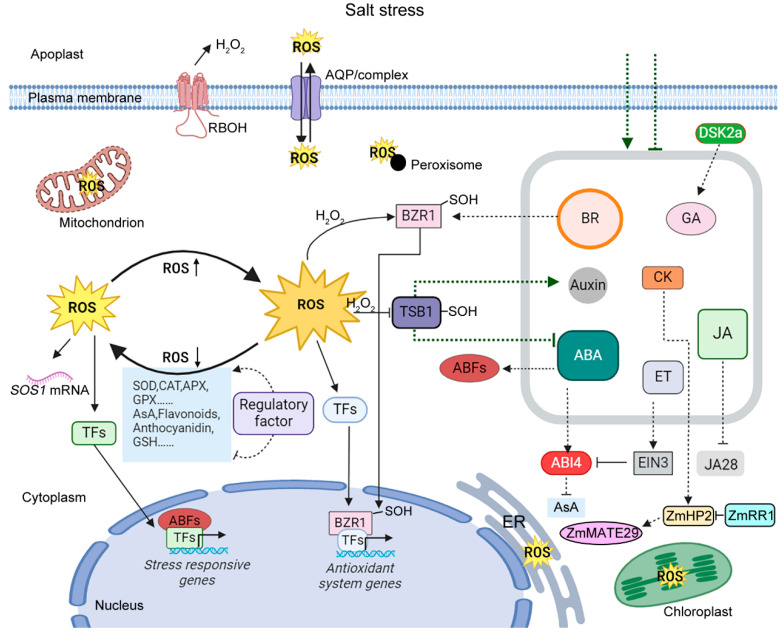
ROS and phytohormones in the context of salt stress. In the early phases of salt stress, RBOHs govern the outburst of ROS in organelles including chloroplasts, mitochondria, peroxisomes, etc. Due to the promotion of bidirectional transportation of H_2_O_2_ by AQP/complexes on the cell membrane and organelle membranes, the level of H_2_O_2_ in the cytoplasm could influence that in other organelles and vice versa. The ROS signal produced by salt stress regulates some transcription factors (TFs), facilitating the expression of some salt stress responsive genes. The stability of *SOS1* mRNA is also enhanced to regulate ionic homeostasis. Persistent salt stress could result in the excessive accumulation of ROS in plant cells when certain TFs are promoted to activate the expression of antioxidant system genes. Antioxidative enzymes, combined with non-enzymatic antioxidants, participate in the control of ROS homeostasis under salt stress. Meanwhile, phytohormone signals are activated or inhibited to regulate salt-stress responses and plant growth. Several regulators involved in phytohormone signals are also controlled by ROS homeostasis, which mediates the crosstalk between ROS and phytohormones under salt stress.

**Figure 2 antioxidants-14-00063-f002:**
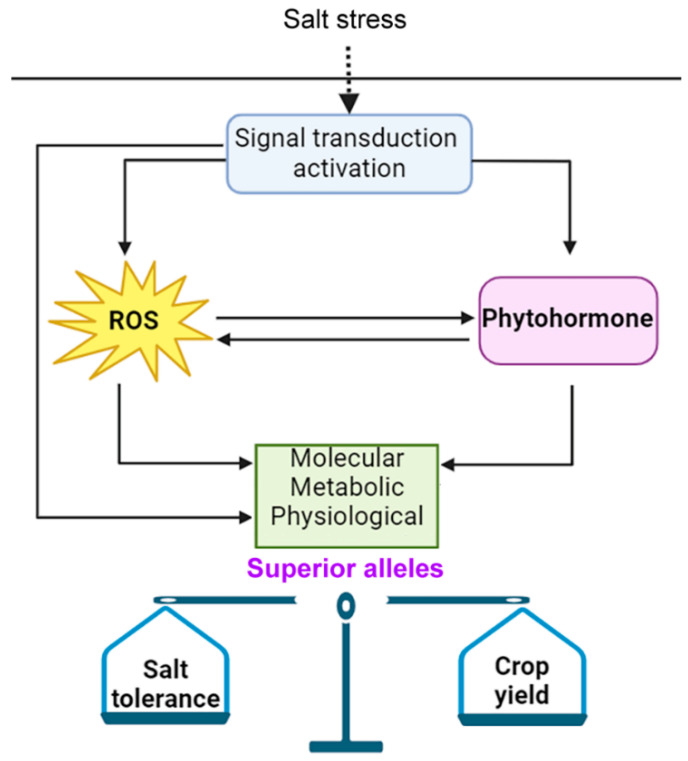
ROS and phytohormones mediate the balance between crop yield and salt tolerance. ROS and phytohormones display an intricate signal crosstalk and jointly govern the growth responses of plants to salt stress. Exploring the key salt-tolerant genes therein and identifying the associated excellent natural variations offer opportunities for striking a balance between salt tolerance and growth.

**Table 1 antioxidants-14-00063-t001:** Genes associated with ROS in crop salt–alkali tolerance.

Gene	ROS	Species	Salt–Alkali Tolerance	Ref.
*Rbohs*	ROS generation	*Oryza sativa*, *Glycine max*	Positive	[17] [20]
*GmNTL1*	ROS generation	*Glycine max*	Positive	[20]
*miR408*	Regulated by ROS signal	*Zea mays*	Positive	[27]
*HSFB2b*	Regulated by ROS signal	*Glycine max*	Positive	[29]
*ZmORF355*	ROS scavenging	*Zea mays*	Positive	[35]
*OsGRF7* *OsUGT1*/*5*	ROS scavenging	*Oryza sativa*	Positive	[36]
*ZmmiR169q*	ROS scavenging	*Zea mays*	Positive	[37]
*ZmNF-YA8* *ZmPER1*	ROS scavenging	*Zea mays*	Positive	[37]
*STRK1*	ROS scavenging	*Oryza sativa*	Positive	[38]
*PC1*	ROS scavenging	*Oryza sativa*	Negative	[39]
*OsVTC1*	ROS scavenging	*Oryza sativa*	Positive	[44]
*ZmBZ1*	ROS scavenging	*Zea mays*	Positive	[45]
*OsBCAT2*	ROS scavenging	*Oryza sativa*	Positive	[46]
*OsGSA1*	ROS scavenging	*Oryza sativa*	Positive	[47]
*TaCHP*	ROS scavenging	*Triticum* *aestivum*	Positive	[48]
*AT1*	ROS scavenging	*Sorghum bicolor* *Oryza sativa* *Zea mays*, *Triticum* *aestivum*	Negative	[49,50]

**Table 2 antioxidants-14-00063-t002:** Genes associated with phytohormones in crop salt tolerance.

Gene	Phytohormone	Species	Salt Tolerance	Ref.
*GmDi19-5*	ABA signal	*Glycine max*	Negative	[56]
*OsSGL*	ABA synthesis	*Oryza sativa*	Negative	[58]
*OsNCED3*	ABA synthesis	*Oryza sativa*	Positive	[58]
*OsSIT1*	ET synthesis	*Oryza sativa*	Negative	[62]
*MPK3/6*	ET synthesis	*Oryza sativa*	Negative	[62]
*OsGA2ox5*	GA metabolism	*Oryza sativa*	Positive	[65]
*OsCYP71D8L*	GA metabolism CK metabolism	*Oryza sativa*	Positive	[66]
*OsDSK2a*	GA metabolism	*Oryza sativa*	Positive	[67]
*JAZ8*	JA signal	*Oryza sativa*	Positive	[70]
*IbNAC087*	JA synthesis	*Ipomoea batatas*	Negative	[72]
*OsARF18*	Auxin signal	*Oryza sativa*	Negative	[73]
ZmRR1	CK signal	*Zea mays*	Negative	[79]
*ZmHP2* *ZmMATE29*	CK signal	*Zea mays*	Positive	[79]

**Table 3 antioxidants-14-00063-t003:** Genes associated with both ROS and phytohormones under salt stress.

Gene	ROS	Phytohormone	Species	Salt Tolerance	Ref.
*BZR1*	ROS scavenging	BR signal ABA synthesis	*Triticum* *Aestivum* L. *Arabidopsis thaliana*	Positive	[80,86]
*ABI4*	ROS scavenging	ABA signal ET signal	*Arabidopsis thaliana*	Negative	[81]
*EIN3*	ROS scavenging	ET signal ABA signal	*Arabidopsis thaliana*	Positive	[81]
*RGA1*	ROS scavenging	GA signal	*Oryza sativa*	Positive	[83]
*OsMYB2*	ROS scavenging	ABA signal	*Oryza sativa*	Positive	[84]
*GPX3*	ROS scavenging	ABA signal	*Arabidopsis thaliana*	Positive	[85]
*TSB1*	Regulated by ROS signal	ABA synthesis, auxin metabolism	*Arabidopsis thaliana*	Positive	[87]
*ITPK4*	ROS scavenging	Auxin metabolism	*Arabidopsis thaliana*	Positive	[88]
*GmSIN1*	ROS generation	ABA synthesis	*Glycine max*	Positive	[89]

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
