# Peer review of "The Regulation of ROS and Phytohormones in Balancing Crop Yield and Salt Tolerance"

_antioxidants, 2025, doi:10.3390/antiox14010063_

Round 1

Reviewer 1 Report

The Regulation of ROS and Phytohormones in the Balance of Crop Yield and Salt Tolerance

Abstract

As a suggestion, in abstract in line 11, change “deteriorated” by “increased”, see comment in MS.

Use as key words significant words but not the ones in title

Introduction

Ok

Materials and methods

As a suggestion add a section of methodology indicating database sources, keywords use in the search of articles, dates. In general, how information was analyzed.

The Regulation of ROS and Phytohormones in the Balance of Crop Yield and Salt Tolerance

Abstract

As a suggestion, in abstract in line 11, change “deteriorated” by “increased”, see comment in MS.

Use as key words significant words but not the ones in title

Introduction

Ok

Materials and methods

As a suggestion add a section of methodology indicating database sources, keywords use in the search of articles, dates. In general, how information was analyzed.

Author Response

As a suggestion, in abstract in line 11, change “deteriorated” by “increased”, see comment in MS.

(R) Thanks for your suggestion. We have changed the word in the abstract.

Use as key words significant words but not the ones in title

(R) Thanks for your suggestion. We have added the key words “reactive oxygen species signaling” and “oxidative stress” in the revised manuscript.

As a suggestion add a section of methodology indicating database sources, keywords use in the search of articles, dates. In general, how information was analyzed.

(R) Thanks for your suggestion. The review does not require a lot of methods, so we added the method of searching sources in the last paragraph of the introduction. We hope it could provide valuable information.

Reviewer 2 Report

The review manuscript is rather well structured and written. I have no many comments. However, to improve the manuscript please pay attention to the sentences like

L91. ...ROS could modify microRNA...

As it is written, it seems it acts directly, and there are more colloquial sentence like this. Even, You have explained in the next sentence or chapter please take care to have all statement clearly and precise. 

Additionally, I suggest to strengthen the parts with explanations on epigenetic mechanisms, there where applied.

I have not found many, but here are some minor remarks

L135. please put the space before OsVTC1-1

L285. sorghum should be Sorghum and in italics

Figures needs to be labelled with summary explanation not just title.

Author Response

Is the quality and presentation of the figures satisfactory?

It can be improved. Explanation of the figures needs to be summarized in the labels.

(R) Thanks for your comments. The explanation of the figures was supplied after the title of the figures in the original manuscript. Maybe due to the formatting reasons, the legends become a part of the main text. We have adjusted it in the revised manuscript.

The review manuscript is rather well structured and written. I have no many comments. However, to improve the manuscript please pay attention to the sentences like L91. ...ROS could modify microRNA...

As it is written, it seems it acts directly, and there are more colloquial sentence like this. Even, you have explained in the next sentence or chapter please take care to have all statement clearly and precise. 

(R) Thanks for your comments. We have corrected it as “It has been discovered that ROS could regulate gene expression through mRNA stability and microRNAs abundance during salt stress responses.” in the revised manuscript. Meanwhile, we have changed some colloquial sentences.

Additionally, I suggest to strengthen the parts with explanations on epigenetic mechanisms, there where applied.

(R) Thanks for your suggestion, we are highly delighted that you have specifically focused on the role of ROS in epigenetic mechanisms. Nevertheless, upon our literature review, it was discovered that although stress-induced oxidative stress can regulate transcription through epigenetic mechanisms, the research on salt stress is relatively scarce. As a result, we can only offer a preliminary summary.

I have not found many, but here are some minor remarks

L135. please put the space before OsVTC1-1

(R) Thanks for your comments. We have corrected it in the revised manuscript.

L285. sorghum should be Sorghum and in italics

(R) Thanks for your comments. We have corrected it in the revised manuscript.

Figures needs to be labelled with summary explanation not just title.

(R) Thanks for your comments. We have adjusted it.

Reviewer 3 Report

the research topic is very interesting and the presented data could contribute to the scientific community. The analysis of the mechanism of how plants react to salt stress is fast developing field with high impact to the environment and humans. 

the introduction is well-organized easy to follow and understandable. the authors point out the importance and influence of potassium and sodium and their concentration on plant metabolism. in addition, they summarize reactive oxidative species and the way of their protective activity in the plant's metabolism.

In the first paragraph, they analyze ROS as having two modes of action, active and passive, with specific plant species for which they are specific.

In the second, under the title . Oxidative Stress Induced by ROS Under Salt Stress, they discuss how ROS can react as an indicator of stress and itself act in plants, causing defects in DNA.

The figures and tables presented in the paper are adequate and the quality of the paper makes it easier to understand and conclude.

The conclusion shows the most important conclusions of the discussion and underlines the possibility of future application of gene correction for better yield in conditions where plants are stressed by salt.

Remarks

in my opinion the abbreviation shouldn't be given in keyword or at least the give full name and abbreviation.

examples of phytohormones that reduce plants growth. 

In the first paragraph, they analyze ROS as having two modes of action, active and passive, with specific plant species for which they are specific.

In the second, under the title . Oxidative Stress Induced by ROS Under Salt Stress, they discuss how ROS can react as an indicator of stress and itself act in plants, causing defects in DNA.

The figures and tables presented in the paper are adequate and the quality of the paper makes it easier to understand and conclude.

Remarks

in my opinion the abbreviation shouldn't be given in keyword or at least the give full name and abbreviation. line 21

examples of phytohormones that reduce plants growth. paragraph 3

The conclusion shows the most important conclusions of the discussion and underlines the possibility of future application of gene correction for better yield in conditions where plants are stressed by salt.

Author Response

(R) Thanks for your comments, which have given us the opportunity to improve the manuscript.

Remarks

in my opinion the abbreviation shouldn't be given in keyword or at least the give full name and abbreviation. line 21

(R) Thanks for your comments. We have changed it.

examples of phytohormones that reduce plants growth. paragraph 3

(R) Thanks for your comments. The phytohormones mentioned in this review have included the ones reducing plant growth, such as ethylene and ABA, which inhibit root growth and seedling growth in rice.

The conclusion shows the most important conclusions of the discussion and underlines the possibility of future application of gene correction for better yield in conditions where plants are stressed by salt.

(R) Thanks for your assessment.

Reviewer 4 Report

In the manuscript by the Jiang et al, the author has outlined recent progress in the molecular regulations underlying crop salt tolerance, focus-16 ing on the double-edged sword effect of ROS, the regulatory role of phytohormone, and the trade-17 off effects of ROS and phytohormone between crop yield and salt tolerance. The manuscript is ok, however it needed a few modifications for its improvement.

1. The title of the manuscript looks confusing, the author needs to modify it as "The Regulation of ROS and Phytohormones in balancing Crop Yield and Salt Tolerance".

2. The abstract is too short, the needs to increase the length of the abstract highlighting the importance of the current review.

3. In the introduction part, the author should elaborate more on what are phytohormones, and why these are important for crops.

4. The author should add a section on mechanism of action of ROS and phytohormones in balancing the crop yield.

5. The future prospects should be elaborated more and the author should highlight the importance of phytohormones in crop yield and salt tolerance.

6. The conclusion section should be elaborated more highlighting the importance of work.

7. More recent references should be cited and discussed.

8. The author can add a figure on the mechanism of action.

In the manuscript by the Jiang et al, the author has outlined recent progress in the molecular regulations underlying crop salt tolerance, focus-16 ing on the double-edged sword effect of ROS, the regulatory role of phytohormone, and the trade-17 off effects of ROS and phytohormone between crop yield and salt tolerance. The manuscript is ok, however it needed a few modifications for its improvement.

1. The title of the manuscript looks confusing, the author needs to modify it as "The Regulation of ROS and Phytohormones in balancing Crop Yield and Salt Tolerance".

2. The abstract is too short, the needs to increase the length of the abstract highlighting the importance of the current review.

3. In the introduction part, the author should elaborate more on what are phytohormones, and why these are important for crops.

4. The author should add a section on mechanism of action of ROS and phytohormones in balancing the crop yield.

5. The future prospects should be elaborated more and the author should highlight the importance of phytohormones in crop yield and salt tolerance.

6. The conclusion section should be elaborated more highlighting the importance of work.

7. More recent references should be cited and discussed.

8. The author can add a figure on the mechanism of action.

Author Response

The title looks confusing, needs modification.

(R) Thanks for your comments. We have changed the title as “The Regulation of ROS and Phytohormones in Balancing Crop Yield and Salt Tolerance”.

The author needs to increase the length of the abstract highlighting the importance of the current review.

(R) Thanks for your comments. We have added some sentences in the revised manuscript.

In the manuscript by the Jiang et al, the author has outlined recent progress in the molecular regulations underlying crop salt tolerance, focusing on the double-edged sword effect of ROS, the regulatory role of phytohormone, and the trade-off effects of ROS and phytohormone between crop yield and salt tolerance. The manuscript is ok, however it needed a few modifications for its improvement.

(R) Thank you very much for your comments. As detailed below, your comments were addressed and alterations in the manuscript were done in the form of tracked changes. We hope the new version of the manuscript is in accordance with the reviewer’s expectations.

  1. The title of the manuscript looks confusing, the author needs to modify it as "The Regulation of ROS and Phytohormones in balancing Crop Yield and Salt Tolerance".

(R) Thanks for your comments. We have changed it.

  1. The abstract is too short, the needs to increase the length of the abstract highlighting the importance of the current review.

(R) Thanks for your comments. Some sentences have been added in the abstract.

  1. In the introduction part, the author should elaborate more on what are phytohormones, and why these are important for crops.

(R) Thanks for your comments. We have elaborated the concept of plant hormones and their significance for crops in the revised manuscript.

  1. The author should add a section on mechanism of action of ROS and phytohormones in balancing the crop yield.

(R) Thanks for your comments. The mechanisms of ROS and phytohormones in coordinately regulating plant growth under salt stress have been unraveled in many species, however, the study of their roles in balancing the crop yield and salt tolerance is limited. Therefore, we didn’t add the section. But the regulation mechanisms of the key genes in crop salt tolerance we searched have been elucidated in Part 4. ROS and Phytohormone Jointly Balance Crop Yield and Salt Tolerance.

  1. The future prospects should be elaborated more and the author should highlight the importance of phytohormones in crop yield and salt tolerance.

(R) Thanks for your comments. We have added the importance of phytohormones in future prospect in the revised manuscript.

  1. The conclusion section should be elaborated more highlighting the importance of work.

(R) Thanks for your comments. We have added it at the beginning of the first paragraph in conclusion section.

  1. More recent references should be cited and discussed.

(R) Accepted, two references reported in 2024 have been added and discussed ( [81] and [94]) in the revised manuscript.

  1. The author can add a figure on the mechanism of action.

(R) Thanks for your suggestion. The mechanism of some recent reported key genes in crop salt tolerance have been shown in Figure 1, which are involved in ROS homeostasis or phytohormone.

Round 2

Reviewer 4 Report

The author makes revisions.

The author makes revisions.